# Sparsity in an artificial neural network predicts beauty: Towards a model of processing-based aesthetics

**Nicolas M. Dibot**[1,2]*, **Sonia Tieo**[1], **Tamra C. Mendelson**[3], **William Puech**[2]⊙, **Julien P. Renoult**[1]⊙

**1** CEFE, Univ. Montpellier, CNRS, EPHE, IRD, Montpellier, France, **2** LIRMM, Univ. Montpellier, CNRS, Montpellier, France, **3** Department of Biological Sciences, University of Maryland, Baltimore County, Baltimore, Maryland, United States of America

⊙ These authors contributed equally to this work.
* nicolas.dibot@cefe.cnrs.fr

**Data Availability Statement:** The code that allowed to obtain the results of this work is available on this Github repository: https://github.com/NicolasDibot/sparsity_beauty.

## Abstract

Generations of scientists have pursued the goal of defining beauty. While early scientists initially focused on objective criteria of beauty ('feature-based aesthetics'), philosophers and artists alike have since proposed that beauty arises from the interaction between the object and the individual who perceives it. The aesthetic theory of fluency formalizes this idea of interaction by proposing that beauty is determined by the efficiency of information processing in the perceiver's brain ('processing-based aesthetics'), and that efficient processing induces a positive aesthetic experience. The theory is supported by numerous psychological results, however, to date there is no quantitative predictive model to test it on a large scale. In this work, we propose to leverage the capacity of deep convolutional neural networks (DCNN) to model the processing of information in the brain by studying the link between beauty and neuronal sparsity, a measure of information processing efficiency. Whether analyzing pictures of faces, figurative or abstract art paintings, neuronal sparsity explains up to 28% of variance in beauty scores, and up to 47% when combined with a feature-based metric. However, we also found that sparsity is either positively or negatively correlated with beauty across the multiple layers of the DCNN. Our quantitative model stresses the importance of considering how information is processed, in addition to the content of that information, when predicting beauty, but also suggests an unexpectedly complex relationship between fluency and beauty.

## Author summary

Developing good predictive models of beauty requires understanding what happens in the brain when we find a person or an artwork beautiful. Recent theories in psychology emphasize the importance of considering how the brain processes features, in addition to the features themselves. Features that are efficiently processed by the brain, such as symmetry, fractality, or naturalness are generally perceived as visually attractive. In this study,

**Funding:** This study was funded by the Agence Nationale de la Recherche (ANR-20-CE02-0005-01) received by JPR and WP, by the National Science Foundation (NSF IOS 2026334) received by TM and JPR, and by the CNRS through the MITI interdisciplinary programs (Programme Interne Blanc MITI 2023.1 - Projet: DEEPCOM-L'intelligence artificielle pour étudier la communication) received by WP. ND and ST received a salary from ANR-20-CE02-0005-01. The funders had no role in study design, data collection and analysis, decision to publish, or preparation of the manuscript.

**Competing interests:** The authors have declared that no competing interests exist.

we leveraged the capacity of artificial intelligence to model information processing in the human brain, to evaluate how the beauty of human faces and artistic paintings can be predicted from the efficiency of the neural code. Our results show that the efficiency of information processing can explain approximately one-third of the perception of beauty and emphasize the importance of considering how information is processed when investigating beauty. Additionally, our use of artificial intelligence demonstrates the potential of this technology to help better understand complex human behaviors.

## Introduction

Understanding and predicting beauty has been a goal of humans for thousands of years. Early studies of beauty aimed to identify characteristics that make objects universally beautiful. Examples of such objective characteristics of beauty include the golden ratio, symmetry, or more complex measures such as fractality [1,2]. Later on, artists and philosophers focused on the subjective aspect of beauty, motivated by the recognition that the taste for the beautiful varies between individuals and even with age within individuals [3,4]. Then, over the XX[th] century, an interaction paradigm gained popularity that accounts for the fact that beauty is simultaneously universal and subjective [5]. Beauty is neither a property of objects nor an idiosyncrasy of the observer, rather it emerges from the interaction between the two [6]. It thus has been argued that the key to understanding beauty hides in the brain mechanisms underlying this interaction.

In psychology, the fluency theory of aesthetics [5] proposes that beauty is determined by how the brain processes information, and that fluency—a subjective sensation of ease in information processing—triggers a pleasurable aesthetic experience [7]. Fluency theory fits the interactionist paradigm very well because information processing is influenced by properties of both the objects and the sensory and cognitive systems that process them [8]. Fluency theory also emphasizes that describing information processing *per se*, in addition to the information being processed, is important to understanding beauty. Overall, the theory can explain remarkably well many, if not most, results of psychology on the phenomenon of beauty. For example, it can account for universal preferences for symmetrical, rounded, highly contrasting, and repeated forms (both through time and space), all of these being fluently processed [5], but it also explains context-dependent pleasure received from an acceleration of information flow (e.g., the 'aha effect' following suspense or in optical illusions [9]).

Fluency has been quantified in various ways, each with advantages but also limitations [10]. Reaction time, for example, is easy to measure but it is only indirectly linked to fluency, being constrained by motor mechanisms. Self-report of subjective ease more closely matches fluency, but it is generally poorly reliable [11]. Characteristics like symmetry and fractality have been often used as metrics of fluency [12]. However, using such characteristics re-embeds fluency into the strictly objectivist paradigm of beauty that dismisses the importance of the perceiver. Modeling beauty using the fluency theory thus requires developing metrics that describe the state of the sensory system or the brain while it is processing information. We qualify such metrics as "processing-based". In contrast, "feature-based" metrics are focused on features, which are quantitative characteristics of objects, measured either directly from the object or as they are perceived. Contrary to processing-based metrics, which are interactive by construction, feature-based metrics can thus be either interactive or objective.

Several authors have argued that fluency could be modeled through the concept of efficiency, which describes the capacity to perform a task with an optimal use of neuronal

resources [8,13]. In neuroscience, efficiency is often measured through the sparsity of the neuronal code. A sparse code is one in which the vast majority of neurons are at rest, and only a few, highly specialized neurons are strongly activated [14]. Using the sparsity of neuronal activations to estimate fluency fits the prediction of Winkielman [15], that "fluent patterns should be represented by more extreme values of activation". Accordingly, a previous study found that sparsity of neuronal activations in a model of the primary visual cortex was positively correlated with the attractiveness of the female face as rated by men [16]. Using a similar approach, Holzleitner et al. [17] found that sparsity was the best predictor of face attractiveness as compared to body mass index, sexual dimorphism, averageness and asymmetry. Sparsity is also negatively correlated with aversiveness. Images of abstract patterns with a lower degree of sparsity are more highly aversive to human subjects [18]. While these previous studies collectively support a link between attractiveness and activation sparsity in the primary visual cortex, to what extent beauty and attractiveness can be explained by activation sparsity as measured in the visual system including, but not limited to the primary visual cortex has so far remained unexplored.

We propose to leverage the capacity of artificial neural networks to model the human visual system to investigate the link between neuronal activation sparsity. Some artificial intelligence architectures, and in particular Deep Convolutional Neural Networks (DCNN) can fulfill this role. DCNNs and the visual cortex have similarities in how they process information [19,20]. Accordingly, patterns of neuronal activations within a DCNN have been shown to predict those recorded in the visual cortex of humans [20–24]. Furthermore, previous studies have shown that metrics summarizing the processing of visual information by DCNNs correlate with self-reported assessment of this processing; for example, the mean activation per layer predicts the perceived complexity of an image [25].

Previous studies in aesthetic science have used DCNNs (for reviews, see [26,27]), for example to recognize artworks and classify artistic paintings by their style (e.g., [28]). DCNNs are also able to predict mean (e.g., [29]) or individual [30] opinion scores of beauty, when trained to do so. Features extracted by DCNNs has permitted the elaboration of feature-based metrics used to unravel the determinants of beauty [31,32]. One study has further shown that artistic and non-artistic images can be distinguished by variance of neuronal activations across layers of a DCNN [26]. Although the authors did not explicitly refer to DCNNs as models of human vision, the variance of neuronal activations describes a state of the visual system while processing an image and is thus a processing-based metric. Variance, however, is only loosely related to sparsity, and has no direct link to processing efficiency [33]. The use of DCNNs to measure processing efficiency and model fluency thus remains to be investigated.

In this work, analyzing images of various objects, we test the hypothesis that their beauty can be predicted from the sparsity of their DCNN activations. We compared results when analyzing various objects (non-artistic portrait photographs, figurative and abstract artistic paintings) and associated judgments related to beauty (beauty itself, attractiveness, negative-positive emotion) in order to qualitatively explore how generalizable the link between sparsity and beauty is. Beauty is ubiquitous, and can potentially describe any stimulus from natural landscapes to abstract representations, the physical aspect of a face or the artistic merit of its representation [34,35]. The fluency theory posits a unity of the experience of beauty across these domains, driven by a common biological mechanism: the efficiency of information processing. Moreover, fluency can mediate the evaluation of beauty in all three brain activities involved in aesthetic experiences: perception, cognition and emotions [36–38]. However, while aesthetic experiences can be both positive or negative (e.g., implying sadness or vertigo; [39], like other authors we consider beauty as a subset of only positive, hedonically marked aesthetic experiences [5]. This distinction helps recognize sing that some artworks, for example, can have high aesthetic value while being not beautiful. We thus predict that high sparsity,

describing elevated fluency, is associated with high scores of beauty across visual domains and judgements of beauty.

## Results

We measured the sparsity of artificial neuron activations triggered by images for which an empirical score of beauty or judgment related to beauty has been obtained by psychological in-lab or online experiments. These images belong to four publicly available datasets and represent photographic portraits (CFD [40] and SCUT-FBP5500 [41] datasets) or artistic paintings (abstract painting: MART [42]; paintings from various styles and epochs: JEN [43] datasets; *see Materials and methods*). We analyzed these four datasets separately. Each image was first processed by the standard DCNN VGG-16 [44] pre-trained on the ImageNet dataset [45]. VGG-16 includes 13 convolutions and 2 fully connected layers. ImageNet is a large dataset of 14 million images depicting about 20,000 categories including people, plants, animals, and human-made objects. Training a DCNN on such a large and varied dataset allows modeling a visual cortex that is not specialized to one specific task, in accordance with neurophysiological data [23,46]. For each image, we extracted the activations for each of the 15 studied layers of the network and calculated the sparsity of the distribution of these activations. Thus, for each image, we have 15 measures of sparsity, one per layer.

### Link between sparsity and beauty

We studied whether beauty could be predicted from the sparsity of neuronal activations, and how this prediction varied with the visual domain (perception of photographic portraits *vs.* artistic paintings).

We investigated the different layers separately because each of them encodes different types of visual information and at different spatial scales. Early convolution layers encode information on simple and localized features (e.g., line segments, localized color contrasts), mid-level layers on moderately complex features like circles, and the last convolution and both fully connected layers more complex information (e.g., entire faces) spanning the whole image. For each layer separately, we estimated the variance in beauty scores (coefficient of determination $R^2$ score) explained by sparsity using a linear regression model. We found a significant $R^2$ score for all layers and all datasets ($p < 10^{-6}$ for all models). However, the variance explained by sparsity varied strongly across both datasets (mean $R^2$ across layers from 2,7% for JEN to 4,8% for CFD, Table 1), but also across layers for a given dataset (e.g., for SCUT-FBP5500 from $R^2 = 0,3\%$ for layer conv4_1 to $R^2 = 17,5\%$ for layer conv1_2; Fig 1). For each dataset, we observed at least one layer with a moderate $R^2$ score (maximum: 11%, 18%, 8% and 9% for CFD, SCUT-FBP5500, JEN, MART, respectively). For both face datasets (CFD and SCUT-FBP5500)

**Table 1. Variance ($R^2$ score) in beauty score explained by the sparsity and principal components of activations in VGG16.** $R^2$ score are calculated between the ground truth and the predicted values of beauty scores by a multivariate Ridge linear regression model including sparsity of one layer (first column, mean of 15 $R^2$ scores, one per layer), sparsity of all layers (second column), principal component (PC) scores (explaining 80% of variance in activations) of one layer (third column: mean of 15 $R^2$ scores, one per layer), the scores of the first three PC of all layers (fourth column), or the scores of the first three PC and sparsity of all layers (fifth column) as predictors. $R^2$ score was calculated using a 10-fold cross validation procedure. The four datasets (rows) were analyzed separately.

| | mean $R^2$ (one model per layer, sparsity only) | $R^2$ (one model for all layers, sparsity only) | mean $R^2$ (one model per layer, PC scores only) | $R^2$ (one model for all layers, PC scores only) | $R^2$ (one model for all layers, PC scores + sparsity) |
|---|---|---|---|---|---|
| **CFD** | 4.8% | 24.7% | 1.6% | 4.2% | 19.8% |
| **SCUT-FBP5500** | 3.9% | 28.3% | 0.8% | 4.2% | 29.1% |
| **MART** | 4.4% | 25.7% | 17.1% | 46.4% | 47.6% |
| **JEN** | 2.7% | 13.9% | 0.7% | 2.9% | 14.2% |

photographic portraits

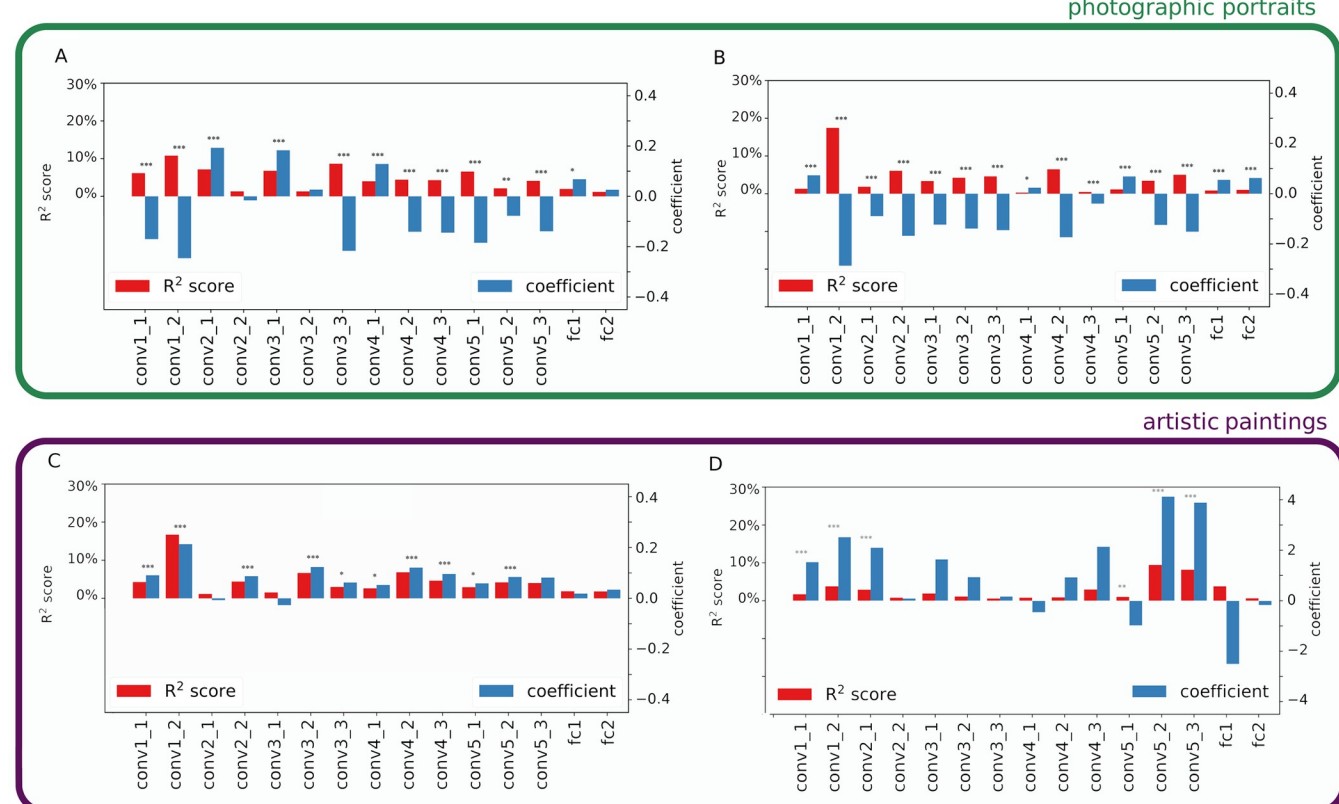

artistic paintings

**Fig 1. Explained variance in beauty score ($R^2$ score) and coefficients of the sparsity of activations for each layer of VGG16.** A: CFD dataset. B: SCUT-FBP5500 dataset. C: MART dataset. D: JEN dataset. Sparsity is measured by the Gini index of the vectorized activations for each convolution and fully connected layers of VGG16 pre-trained on ImageNet. Coefficients were calculated by fitting univariate linear regression models fitted with a 10-fold cross-validation, and the $R^2$ score was calculated between the ground truth and the predicted values. One, two or three asterisks indicate p-value smaller than 0.05, 0.01 and 0.001, respectively.

and for the abstract paintings dataset (MART), the $R^2$ score was the highest for the second convolution layer (conv1_2). For the figurative paintings dataset (JEN), the $R^2$ score was the highest for the two last convolution layers. This means that for faces and abstract paintings (CFD, SCUT-FBP5500, MART), people's perception of beauty is most influenced by the efficiency of processing texture or local color contrasts, while for figurative paintings, the efficiency of processing objects as a whole, and the spatial arrangement of their parts, is more important.

We also examined the coefficients estimated by this multivariate model (blue bars in Fig 1). Some of the layers with a significant effect of sparsity have positive coefficients while others have negative ones. Importantly, even within a broad portion of the network (early *vs*. mid *vs*. deep layers), the sign can vary from positive to negative, especially for face datasets (CFD, SCUT-FBP5500). This puzzling result indicates that from one layer to the next, high scores of beauty could be explained by either very efficient or inefficient neural codes.

We then investigated the link between beauty and the efficiency of information processing across the whole visual system. Indeed, previous studies in psychology have provided evidence that the ease of processing information at each stage of the visual system (e.g., as measured by detection time in the early stages, and as recognition performance in the later stages) triggers micro-experiences of fluency associated to each of these stages, and that these micro-experiences aggregate into one global sensation of fluency [5,47].

To model the aggregation of processing efficiency at every layer, we fitted a single multivariate (one measure of sparsity per layer) linear regression model to beauty scores. Because the model includes fifteen non-independent predictors (see S1 Fig), we constrained the optimization by applying a L2-norm penalty to coefficient estimates (i.e., Ridge regression model). We calculated the adjusted $R^2$ score using a 10-fold cross validation procedure. We found a significant ($p < 10^{-6}$) $R^2$ score for all four datasets. Moreover, sparsity explained approximately 25% of the variance in beauty scores for the two photographic portrait databases (CFD, SCUT-FBP5500) and for the abstract painting database (MART; Table 1, column "$R^2$ (one model for all layers, sparsity only)"). With figurative paintings, sparsity explained a substantial but lower fraction of variance in beauty scores ($R^2 = 14\%$ for JEN; Table 1, column "$R^2$ (one model for all layers, sparsity only)"). Eventually, although a simple measure of sparsity averaged over DCNN layers does not explain much of the variation in beauty (Table 1, column "mean $R^2$ (one model per layer, sparsity only)"), a multivariate model including all layers predicts it with relatively high $R^2$ scores. Yet, postulating that the multivariate regression models how fluency micro-experiences aggregate, our results show that this aggregation could be more complex than the mere sum of micro-experiences and may rather involve top-down controls weighting their importance in different stages of information processing.

## Comparison with a feature-based model

In order to compare our processing-based approach to predict beauty to a traditional feature-based approach, we studied to which extent the neuronal activations themselves, rather than the sparsity of their distribution, can explain variation in beauty scores (following the method described by Iigaya [30]). In a DCNN, each artificial neuron describes one feature in a given region of the image, through its weighted connections to previous neurons. The magnitude of its activation indicates the importance of that specific feature in that region of the image. We then used neuronal activations as predictors in multivariate linear ridge regression models, fitting one model per layer at a time. However, because there were too many activations for fitting the models (the first convolution layer includes 3,211,264 activations), we reduced the number of activations using a principal component analysis (PCA), keeping only those components that explain a total of 80% of variance in activations (between 125 and 2,162 principal components depending on layers and datasets; see Supp Inf.). For all layers and all datasets, the $R^2$ score was significant ($p < 10^{-6}$ for all models), indicating that features, from simple and localized to complex and holistic, do explain variation in beauty scores for both face images and artistic paintings. However, activations explained a low fraction of variance in beauty scores, with $R^2$ score never exceeding 4% (Fig 2). One major exception is for the abstract painting dataset MART, with $R^2$ score ranging from 6% to 42%.

To analyze how encoded features could predict beauty at the entire network level, as for sparsity we used a single multivariate Ridge regression model using the PC scores of all layers. However, to keep the model tractable, we considered only the first three components of each layer. Again, except for MART, we found that the first three principal components explain a relatively low fraction of variance in beauty ($R^2$ score < 5%; Table 1, column "$R^2$ (one model for all layers, PC scores only)"), in line with the previous result obtained when using all principal components that explain 80% of the variance in activations.

Last, we analyzed the benefits of combining both feature-based and processing-based metrics to predict beauty by fitting one multivariate ridge regression model per dataset, including the first three principal components of the PCA applied to activations plus sparsity for each layer, and considering all layers in the same model (i.e., 60 predictors). The $R^2$ score was always significant ($p < 10^{-6}$) and varied from 14% for the figurative paintings (JEN dataset) to up to

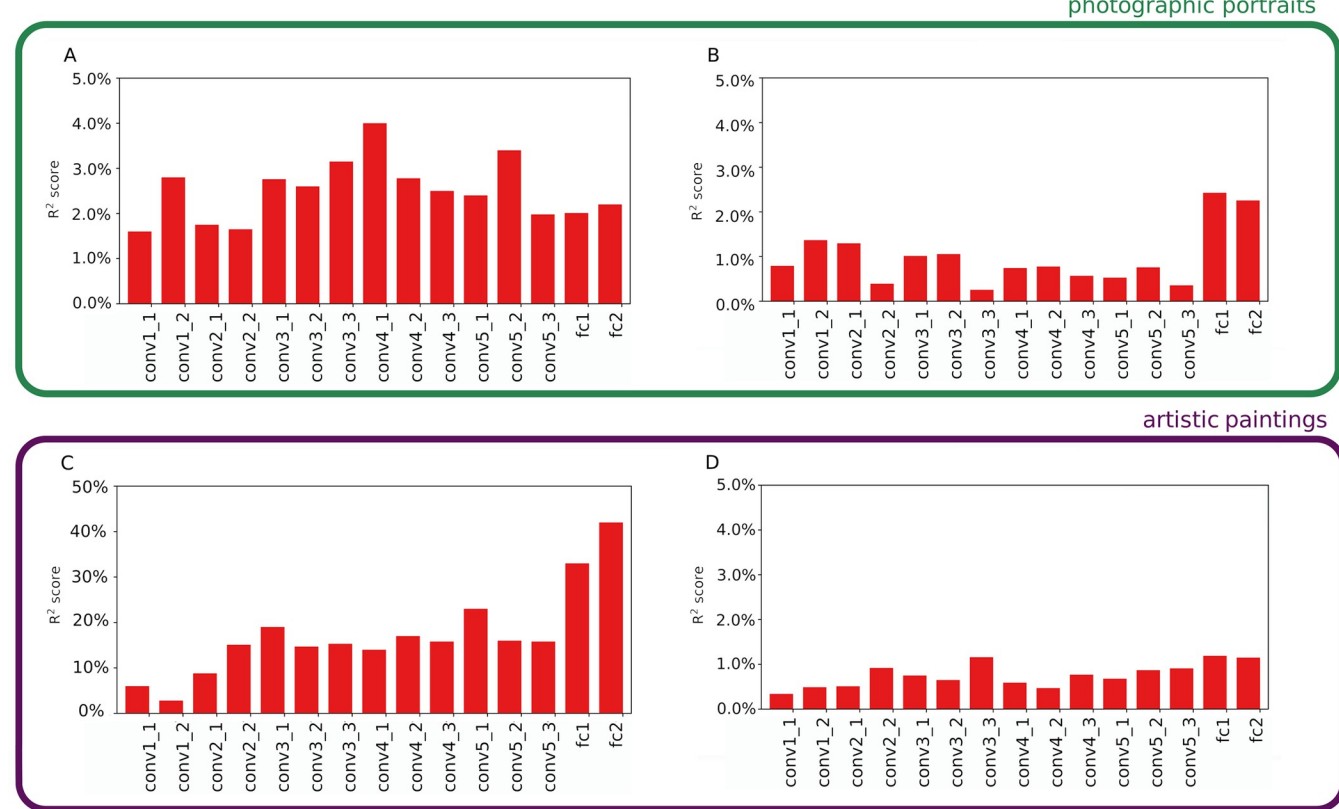

**Fig 2. Variance in beauty score ($R^2$ score) explained by the principal components of activations in different layers of VGG16.** A: CFD dataset. B: SCUT-FBP5500 dataset. C: MART dataset. D: JEN dataset. The principal components are those explaining 80% of the total variance of the activations for each convolution and fully connected layers of VGG16 pre-trained on ImageNet. The $R^2$ score was calculated between the ground truth and the predicted values by a multiple ridge linear regression model fitted with a 10-fold cross-validation procedure.

48% for abstract paintings (Table 1, column "$R^2$ (one model for all layers, PC scores + sparsity)"). Comparing these results to those with the model including sparsity only (Table 1, column "mean $R^2$ (one model per layer, sparsity only)"), the increase in explained variance from adding the feature-based metrics is marginal (SCUT-FBP5500: +0.8%; JEN: +0.2%), or even negative (CFD: -4.9%). Only for the abstract paintings, considering features in addition to processing efficiency significantly increased the ability of the model to predict beauty scores (MART: +19.6%). Comparing these results to those with the model including PC scores only (Table 1, column" $R^2$ (one model for all layers, PC scores only)"), considering sparsity increased the predictive capacities of the model by a factor 5 to 7. For the abstract paintings, considering sparsity only marginally increased the explained variance (MART: +1.2%).

## Discussion

It has been widely acknowledged that beauty results from the interaction between a stimulus and a beholder, however, the neuronal mechanisms describing this particular interaction have remained unknown. Here, we hypothesized that the way information is processed (processing-based aesthetics), more than the type of information being processed (feature-based aesthetics), drives the evaluation of beauty. We thus investigated whether visual beauty could be predicted from the sparsity of neuronal activations in an artificial deep convolutional neural network (DCNN), a statistic characterizing the efficiency of processing visual stimuli that is

independent of the features describing these stimuli. Our results clearly demonstrate the importance of considering the efficiency of information processing to predict beauty. Moreover, we found that for some categories of stimuli (faces and figurative artistic paintings), information processing itself likely contributes more than the processed features for explaining variation in perceived beauty. However, we also found that processing efficiency was either positively or negatively correlated with beauty, depending on which layer of the DCNN was considered.

Although demonstrated here with the help of artificial intelligence, the idea that the processing of stimuli *per se* is an important determinant of beauty is an old one in the field of aesthetics. The German philosopher Immanuel Kant, for example, proposed that beauty is not contained in the object, but is rather an effect of the state of mind of the subject, brought about by the object [48]. The theorist of literature Gérard Genette also famously claimed that "it is not the object that makes the relation aesthetic but the relation that makes the object aesthetic" [49]. Our results, showing that sparsity can explain variation in beauty across different visual domains and components of beauty (i.e when evaluating the physical beauty of faces, or the artistic beauty of abstract or figurative paintings), offer strong support to the interactionist paradigm of beauty and suggest a common biological mechanism underlying all evaluations of beauty.

One previous study proposed to quantify fluency through statistical typicality [50], based on observations that more typical stimuli are also more easily processed [51]. Statistical typicality is related to the psychological concept of familiarity, the manipulation of which has been pivotal in supporting the fluency theory of aesthetics (e.g., [36]). Typicality was measured as the likelihood of a stimulus in the latent space of an encoder, given a reference distribution. Using CFD as the reference distribution, the authors found that facial typicality explained 15% of variance in facial attractiveness. Crucially, sparsity also captures the concept of statistical typicality because, during training, the weights of a DCNN are tuned to the statistical distribution of features of the training dataset (in our case, ImageNet), such that images with a similar distribution are sparsely encoded during inference [52]. However, while typicality specifically refers to familiarity, sparsity more explicitly characterizes the use of neuronal resources, that is, the efficiency of information processing *per se*, a presumably important component of fluency [8]. Regardless of the level of overlap between typicality and sparsity, both [34] and this study considered a static perceptual system. In contrast, a classical approach to study fluency in psychology is to investigate how the system varies when manipulating fluency, for example using matching prime (see [36]). In the future, it would thus be interesting to apply similar dynamic approaches to DCNNs, for example by analyzing how progressively tuning the DCNN to features that are specific to the target stimuli increases the ability of sparsity to explain beauty.

Comparing the predictive capacity of a model including sparsity only with a model including both sparsity and features suggests that information processing could be more important than features in determining beauty. One exception is for the abstract paintings MART dataset, for which features explained more variation in beauty scores than sparsity did, and sparsity did not improve prediction. The fact that sparsity did not significantly improve the model with PC scores only, while it could alone explain 25.7% of variance of beauty scores, suggests that, for abstract paintings, most of the information conveyed by processing efficiency is redundant with information conveyed by features. Given that the negative-positive emotions were evaluated in the abstract paintings dataset, one explanation to the difference between this dataset and other datasets could be that the hedonic marking of efficient information processing in general is not mediating the link between beauty and emotions. This, however, contradicts previous accounts suggesting at least a mild effect of fluency on aesthetic emotions (for a

review, see [53]). For this dataset, beauty was scored by lay people who thus based their judgment on the design of stimuli only, and not on representations encoded in as in figurative paintings. Our result could thus rather suggest that information processing influences aesthetics only when stimuli elicit interpretation or meaning assignment (see also [54]).

The relative importance of sparsity and features in predicting beauty should be interpreted with caution, however. Indeed, the principal component analysis is a commonly used method to describe features (*e.g.*, [30,55]), but it is reductive by construction. Unfortunately, accounting for the full variation in features inevitably poses limitations, either technical or biological. The use of a PCA to reduce the dimensionality of the feature space is an example of technical limitation imposed by statistical regression modeling. To circumvent this limitation, other studies have trained DCNN to directly predict beauty scores, obtaining remarkably high $R^2$ scores (up to 90% in [56]). While these studies highlight that features can theoretically encode most of the information about beauty, in practice, they are not biologically realistic because, contrary to DCNN, the visual cortex as a whole has been shaped by evolution and development to perform many tasks, not to predict beauty only. More work is clearly needed both to quantify the relative importance of processing fluency and stimulus features in predicting beauty, and to understand the factors influencing this relative importance.

Interestingly, for a given dataset, the most predictive layers differ when considering either information processing or features. For MART and SCUT-FBP5500, for example, beauty is best predicted by the last two layers, Fc1 and Fc2 (describe whole-image features) when considering PCA features, and by the first convolutional layers (describe textures and local color contrasts) when considering sparsity. The dissociation between processing-based and feature-based contributions to beauty highlights the richness of the aesthetic experience and myriad possibilities that naturally or culturally shaped communication signals have to increase their attractiveness.

Although our results validate our main hypothesis that beauty can be explained in part from information processing, they reject the prediction that high sparsity is associated with high scores of beauty. Indeed, we found that depending on layers and datasets, high beauty scores could be associated with either highly efficient, or the opposite, highly energy-demanding coding (see Fig 1). This result contradicts the existence of a universal positive correlation between processing efficiency and beauty, and thus the original (and most frequently cited) formulation of the fluency theory of aesthetics. Yet, a recent study in psychology has also found that beauty could be negatively correlated with fluency [57]. One explanation is that fluency would work as an amplifier, being positively correlated with beauty for stimuli with a positive emotional valence, and negatively correlated for negatively-valenced stimuli [58]. This is in line with the general idea that a core goal of perceptual processing is to disambiguate stimuli and thus enable the most appropriate behavior [59]. Another explanation is that relative, rather absolute fluency, determines beauty [9], and thus that alternating between low and high fluency contributes to increasing beauty [60]. With suspense, or optical illusions, for example, pleasure arises from a sudden increase in information processing that was purposely blocked to amplify the feeling of fluency. In music, alternating between consonant and dissonant chord structures contributes to the pleasure of listening and exemplifies the more general tension-resolution hypothesis of aesthetic experience [61]. Whether similar phenomena occurring between processing stages within the information pathway is necessary to trigger an aesthetic experience remains to be investigated, but our data from artificial neural networks suggest that these hypotheses deserve investigation. In any case, when considering the different stages of visual perception, fluency appears to be linked to beauty in more complex ways than previously thought.

Finally, our study relies on both assumptions that DCNNs can model information processing, and that they are valuable tools to study complex mental phenomena. The first assumption has remained controversial [62] as several studies have highlighted differences, some of them significant, between DCNNs and biological vision in the underlying computation (e.g. [63,64]). For example, DCNNs rely on texture information more than shape in object recognition [65]. Yet for many neuroscientists, the emergence in DCNNs of the most fundamental properties observed in animal vision indicates that they are still powerful tools for modelling and understanding this perceptual modality [66,67]. Regarding the second assumption, we acknowledge that purely feed-forward models like VGG16 fall short in describing the full complexity of the judgment of beauty. However, despite top-down cognitive controls that likely influence this judgment, several studies have stressed that it can be made independently of any appeal to cognition [68–70]. Like previous studies showing that VGG16 models reasonably well how the human brain generally processes visual information (e.g., [19]), we concur that the simplicity and tractability of this DCNN can reveal overlooked important brain processes. In particular, it can open a new era in the centuries-old quest to explain and predict beauty.

## Materials and methods

### Datasets

We used four different, publicly available datasets consisting of images and associated mean opinion scores (MOS) of beauty (Table 2). The Chicago Face Database (CFD v3.0) includes 827 standardized photographic portraits (centered faces, identical outfits, identical camera settings) of American and Indian adults of both genders, from various self-reported ethnic origins [40]. Image size is 2,444x1,718 pixels. For each image, a MOS of attractiveness was obtained by averaging ratings from 1,087 American participants, along a Likert scale from 1 to 7, with 1 being not attractive at all and 7 extremely attractive. (Question R013 from CFD Codebook). See S2 Table for more information on the evaluators.

The SCUT-FBP5500 database (South China University of Technology—Facial Beauty Prediction) includes 5,500 photographic portraits of Asian and Caucasian adults of both genders [41]. Portrait images were retrieved from various sources and thus are not standardized. Image size is 350x350 pixels. Beauty MOS were obtained by averaging ratings from 60 Chinese volunteers, along a Likert scale from 1 to 5 with 1 being the least attractive and 5 the most attractive. See S2 Table for more information on the evaluators.

The MART database (Museum of Modern and Contemporary Art of Trento and Rovereto) includes images of 500 abstract paintings by different artists from Trento museum in Italy [42]. The width and height of images varies from 59 to 812, and 45 to 1,036 pixels, respectively. One hundred Italian laypersons were asked to rate their emotional response to each image along a Likert scale from 1 to 7, with 1 being the most negative and 7 the most positive. Previous studies have shown that beauty is mostly associated with positive emotions (e.g., [71]). See S2 Table for more information on the evaluators.

**Table 2. Description of image databases associated with a beauty score.**

| Name | # Images | Type | Evaluated component of beauty |
|---|---|---|---|
| CFD | 827 | Portraits | Attractiveness |
| SCUT-FBP5500 | 5,500 | Portraits | Beauty |
| MART | 500 | Abstract paintings | Negative-positive emotion |
| JEN | 1,563 | Representational paintings | Beauty |

Last, the JEN database (from the German town of Jena) contains 1,563 figurative artistic paintings by different artists representing different movements [43]. Image size is 1,456–30,000x1,351–23,803 pixels. For each image, beauty was evaluated by 134 observers along a Likert scale from 0 to 100, with 0 being not beautiful and 100 beautiful. See S2 Table for more information on the evaluators.

For all analyses, images were resampled to 224x224 pixels (input size of VGG16), and scores of beauty or their proxy were all standardized to the same scale varying between 0 (lowest attractiveness/beauty score) and 5 (highest attractiveness/beauty score).

## Encoding images with a convolutional neural network

All images were encoded using the VGG16 [44] deep convolutional neural network pretrained on the ImageNet dataset [45]. VGG16 includes 13 convolution layers and two fully connected layers. For each image, we thus extracted 15 encodings corresponding to the neuronal activations (after ReLU transformation) of these 15 different layers. For convolution layers, encodings are three-dimensional matrices of size $H*W*C$, with $H$, $W$ corresponding to the height and the width of the feature maps, respectively, and $C$ to the number of feature maps (i.e., channels). $H$ and $W$ vary from 224 to 14, and $C$ from 64 to 512, between the first and the last convolution layer. For both fully connected layers, encodings are one-dimensional vectors of size 4,096.

## Measuring sparsity

The efficiency of image processing at a given network layer was estimated as the activation sparsity, which measures inequity in the distribution of activations in this layer. One study evaluated the ability of various metrics of sparsity to satisfy the attributes desired to properly measure inequity of distribution [72]. The authors found that only the Gini index meets all the desired attributes. For example, kurtosis and $L_1$-norm, two other commonly used metrics of sparsity, fail to predict an increase in inequity when adding null values and extremely high values, respectively. The authors did not include the Treves-Rolls metric [73] in their comparison, despite its popular use in neuroscience. We thus used the Gini index and the Treves-Rolls metrics to measure sparsity. The two metrics yielded qualitatively similar results, thus only results with the Gini index are presented here.

For convolution layers, the $H*W*C$ matrices of activations were first flattened into one-dimensional vectors. These vectors, and the vectors of fully connected layers were sorted in ascending order. The Gini index was calculated as:

$$G = \frac{\sum_{i=1}^{n}(2i - n - 1)x_i}{n\sum_{i=1}^{n} x_i}$$

with **n** the number of activations in the layer (i.e., vector length), **i** the index and $x_i$ the activation at the index. Image encodings and measurements of sparsity were performed using Python 3.9.5.

## Statistical analyses

The four datasets were studied separately. For each layer, measurements of sparsity were z-transformed prior to statistical analyses. We first conducted simple linear regression models to analyze the correlation between the empirical scores of beauty (response variable) and the sparsity measured for each layer (one model per layer). Then, we analyzed the ability to predict beauty from sparsity measurements of all layers with a single multivariate linear regression

model and a Ridge penalization, that included 15 explanatory variables (a separate sparsity measurement for each of the 15 layers). The strength of the Ridge penalization (Lambda parameter) was set by cross-validation.

Last, we used linear Ridge regression models to analyze the ability of image features to predict beauty scores. However, encodings have too many dimensions to be included in a regression model. For example, in the first convolution layer, the encoding corresponds to $H*W*C =$ 3,211,264 features. We thus first reduced the dimensionality of encodings by applying a PCA (one PCA per layer per dataset), keeping the principal components explaining 80% of variance (see S1 Table). PC (principal components) scores were then z-transformed and included in the ridge regression model as explanatory variables. We performed one Ridge regression for each layer separately. In addition, we performed one global Ridge regression including the first three principal components of each layer (model with 45 explanatory variables), and another global Ridge regression including the first three principal components and the sparsity measurement of each layer (model with 60 explanatory variables). For all linear regression and linear Ridge models, we performed a 10-fold cross-validation, repeated 100 times to ensure that the results were stable, calculating the coefficient of determination ($R^2$ score for univariate models and adjusted $R^2$ score for multivariate models) between predicted and empirical beauty scores for each test fold, and then averaging the $R^2$ score over the 10 folds and then over the 100 repetitions. Statistical analyses were performed using R software v4.2.1 [74]. We used the *glmnet* method of package *glmnet* [75] for the Ridge regression models, and the *caret* package for the cross validation [76].

## Supporting information

**S1 Fig. Pearson correlation matrix of the sparsity of activations in different layers of VGG16.** A: CFD dataset. B: SCUT-FBP5500 dataset. C: MART dataset. D: JEN dataset.
(TIF)

**S2 Fig. Beauty score explained by the Gini index for the layer with the highest $R^2$ score per database.** A: CFD dataset (second convolution layer of the first block). B: SCUT-FBP5500 dataset (second convolution layer of the first block). C: MART dataset (second convolution layer of the first block). D: JEN dataset (second convolution layer of the fifth block). The blue trend line corresponds to the predictions provided by the model.
(TIF)

**S1 Table. Number of PCA components that explained 80% of the total variance of activations for each database and each layer of VGG16.** A: CFD dataset. B: SCUT-FBP5500 dataset. C: MART dataset. D: JEN dataset.
(DOCX)

**S2 Table. Description of evaluators from image datasets.**
(DOCX)

## Acknowledgments

We are grateful to Melvin Bardin for his help with high-performance computing.

## Author Contributions

**Conceptualization:** Nicolas M. Dibot, Tamra C. Mendelson, Julien P. Renoult.

**Formal analysis:** Nicolas M. Dibot.

**Investigation:** Julien P. Renoult.

**Methodology:** Nicolas M. Dibot, Sonia Tieo, Julien P. Renoult.

**Project administration:** Julien P. Renoult.

**Supervision:** William Puech, Julien P. Renoult.

**Writing – original draft:** Nicolas M. Dibot, Julien P. Renoult.

**Writing – review & editing:** Nicolas M. Dibot, Sonia Tieo, Tamra C. Mendelson, William Puech, Julien P. Renoult.

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
