## [Decision Letter · Decision Letter 0]

10 Jul 2023

Dear Mr. Dibot,

Thank you very much for submitting your manuscript "Sparsity in an artificial neural network predicts beauty: towards a model of processing-based aesthetics" for consideration at PLOS Computational Biology.

As with all papers reviewed by the journal, your manuscript was reviewed by members of the editorial board and by several independent reviewers. In light of the reviews (below this email), we would like to invite the resubmission of a significantly-revised version that takes into account the reviewers' comments.

We cannot make any decision about publication until we have seen the revised manuscript and your response to the reviewers' comments. Your revised manuscript is also likely to be sent to reviewers for further evaluation.

Sincerely,

Roland W. Fleming, PhD

Academic Editor

PLOS Computational Biology

Natalia Komarova

Section Editor

PLOS Computational Biology

Reviewer's Responses to Questions

**Comments to the Authors:**

Reviewer #1: I have reviewed the manuscript titled "Sparsity in an Artificial Neural

Network Predicts Beauty: Towards a Model of Processing-Based Aesthetics" by

Dibot and co-authors, submitted for publication in PLOS Computational Biology.

In this study, the authors present an intriguing investigation on the

relationship between the degree of inequality in the hidden layers of a

pre-trained convolutional neural network (VGG-16) and the mean opinion scores

of beauty in four datasets (two datasets of faces and two datasets of

paintings). Specifically, they calculate the Gini index of the activation maps

(after RELU) for each image and correlate these values with the beauty

measures. When considering individual layers, the R² scores range from

approximately 3% to 6%, but considering all layers collectively improves these

values to approximately 14% to 28%. Furthermore, the authors conduct

regression analyses of principal components associated with activation maps,

revealing lower R² scores compared to those obtained with the Gini index, with

the exception of abstract paintings. In addition, they explore the combination

of principal components with the Gini index, finding no significant

improvement in R² scores. The authors adeptly discuss all these findings in

the context of the aesthetic theory of fluency.

I thoroughly enjoyed reading this work and believe it deserves publication in

PLOS Computational Biology. However, there are a few points I would like the

authors to consider before my final recommendation.

- The notion that convolutional neural networks accurately emulate the visual

representations in the human brain is a subject of debate, and I suggest

that the authors acknowledge this limitation of their study.

- Alongside AlexNet, VGG16 is one of the initial deep learning models for

image classification. Hence, I am curious whether the authors have

experimented with different models in their study. Keras provides various

pre-trained models (https://keras.io/api/applications/), and I believe that

considering different models would strengthen the authors' conclusions.

- The sentence "at least one layer with a moderate to high R² score" followed

by the numbers in brackets might surprise most readers. And that made me

wonder about the nature of the association between the GINI and the beauty

scores. Is it possible to include at least one example of this association

(maybe in SI)? Additionally, because beauty scores are very discrete in some

datasets, would it make sense to frame the association between sparsity and

beauty as a classification problem (maybe for future studies)?

- I must confess that I was unaware of the utilization of the Gini index as a

measure of sparsity. I was further surprised when reading Reference [49] and

learning the numerous advantages of this measure. Hence, I suggest that the

authors provide more contextual information to justify their excellent

choice.

- On a minor note, please replace "R2" with "R²".

Reviewer #2: This study investigates the link between diverse subjective scores ("beauty" scores) for images and the sparsity of their representation in a conventional VGG-16 deep neural network that was trained for object recognition on the ImageNet dataset. Specifically, the authors study how much of the variance in the subjective scores can be explained by the sparsity of feature responses of different layers of the neural network. To measure the sparsity of images of faces (2 datasets), of abstract paintings (1 dataset) and figurative paintings (1 dataset), the Gini coefficient is calculated for each layer separately and for all layers. For this purpose, appropriate single and multiple regression models are developed. Results show positive and negative correlations between the ratings and the coefficient. R2 values varied widely between and within datasets. Overall, there is no clear-cut answer to the initial question of what the relation between ratings and sparsity might be, except to say that such correlations can indeed be found.

The study is interesting, timely and the analysis is valid overall (however, see Point 5 below). I do see some weakness in the initial hypothesis and in the interpretation of the data. Given the ambiguous results, some of the interpretations seem overelaborate and a bit fuzzy.

Major points

(1) Line 163-164, "empirical score of beauty": The authors use subjective ratings from four different previous studies. They summarily refer to these ratings as "beauty" ratings. A justification of this usage of the terms is needed. (i) In the two datasets of face images, face attractiveness has been rated, which is fine and well-investigated. (ii) In the MART dataset, positive/negative emotions were rated. How emotions relate to beauty perception is not explained in any detail. Paintings that evoke negative feelings, such as sadness or disgust, can nevertheless be beautiful. Can the authors justify their interpretation of emotional ratings as beauty ratings? (iii) For the JenAesthetics dataset, both ratings of "beautiful" and „aesthetic quality" are available. The authors apparently chose to analyze the "aesthetic" ratings, but they refer to them as beauty ratings. Can the authors give a rationale for this? Moreover, the beauty of a painting (i.e. of the formal image composition) is not the same as the attractiveness of faces depicted in an image (see, e.g., Schulz et al., 2017, doi.org/10.3389/fpsyg.2017.02254). There are famous examples of ugly (unattractive) faces depicted in beautiful paintings (e.g., "The Ugly Duchess" by Quentin Matsys). Given these discrepancies, I wonder whether the ratings from the four datasets can really be compared in terms of "beauty" ratings at all. This issue is important because different ratings depend differentially on image properties in general. The heterogeneity of the rating terms may be one of the reasons for the inconsistent results obtained for the four datasets.

(2) Fluent/sparse/efficient coding, lines 130-142: According to the authors, fluency can be modeled trough the concept of efficiency, which is an interesting idea. They state that "a sparse code in one in which the vast majority of neurons are at rest and only a few … neurons are strongly activated". The authors go on to cite studies showing that high sparsity is associated with high attractiveness/beauty of faces. From the studies cited by the authors, the hypothesis would be that higher sparsity is associated with higher beauty of the stimulus. However, a very sparsely coded stimulus would be an empty white sheet with just one short line on it. Would this stimulus be highly beautiful? If this does not fit the hypothesis, please explain more clearly.

Line 333-334: "beauty is determined mostly by ease of information processing" (i.e., fluency). Elsewhere, fluency is operationalized as sparse/efficient processing. The results from the present study (negative coefficients in Figure 1) directly contradict this central hypothesis, at least in part. Does this result contradict the fluency theory of beauty perception (see also references 41 and 42)?

In conclusion, what I am missing is a clear hypothesis or experimental question that relates to previous research in the field and is tested in this research. With an open question at the beginning (lines 156-158) and ambiguous results at the end, it is not easy to understand what we can learn from this study about the link between beauty, sparsity and efficient coding in the visual system (other than that the relation is a complex one).

(3) The review of previous work by other researchers deserves some more attention, especially in the Introduction section. For example, deep neural networks or CNNs have been used extensively before to predict subjective ratings of beauty, in particular, of faces and photographs, and this should be acknowledged in the Introduction, together with appropriate references. The idea to characterize artworks with low-level CNNs features is also not novel (for example, Brachmann et al., Front. Psychol. 2017, 8:830; Iigaya et al., 2021; see general review on AI and art by Cetinic and She, 2022, doi.org/10.1145/3475799). Moreover, a link between sparse (efficient) coding and beauty perception has been postulated before (see the references in previous papers by the senior author on this topic). Lines 102-109: Insert reference to S. Zeki, a prominent proponent of this idea.

(4) Lines 127-129 and 320-322. "Processing-based metrics … describing the dynamics of information processing.": I am not sure whether I understand this point correctly. What does "dynamic" refer to in terms of information processing? To this reviewer, "dynamic" implies a change or the production of movement of some sort. With the exception of the "aha effect" (lines 118-120), none of the analyses in this manuscript are carried out as a function of time or imply change, but they refer to static properties extracted from the stimuli and models at one point in time. Also, the ratings analyzed do not change over time in the present study. Please clarify what is "dynamic" about your analyses. Also, the difference between "feature-based" aesthetics and "processing-based" aesthetics is not evident to this reviewer. Perceptual features are processed by the visual system, too. What does the low-level visual system process other than image features? Like variance (Brachmann et al., 2017), sparsity is a measure of the global distribution of activation in the network layers. Perhaps derive a term from that definition somehow?

(5) Lines 501-505: The authors carried out 10-fold cross-validation to obtain R2 scores. With small sample size, these scores can be unstable potentially. The authors should consider repeating the cross-validations many times randomly and then calculate and compare the averages obtained from the n-times 10-fold cross-validations. The number of repetitions n should be high enough for testing whether the means are significantly different from 0. This is especially important for data with small R2 values and from small datasets (e.g., the JenAesthetics dataset). If you already did this or something similar, please clarify.

(6) The authors use Gini's coefficient as a measure for uneven distribution or inequality between the measured activations of the CNN features, which is an interesting and valuable idea. Another measure of inequality of values would be variance. Brachmann et al. (2017) studied different types of variance of CNN filters previously. Their results suggest that traditional artworks are richer (i.e., less sparse!) than various types of non-art images at low CNN levels. This result is opposite to the authors' hypothesis of a positive correlation between sparseness/fluency and beauty. The authors may consider discussing this contradiction.

(7) The beginning of the Discussion section (lines 318-356) is too long and has review-like passages. It should be shortened considerably. By contrast, the rest of the Discussion (lines 357-418) focuses on what has actually been found (or not) in the present study and interprets the findings in relation to previous studies. This part of the Discussion provides a much more balanced view of the different theories in the field than the Introduction section. Consider moving some of this into the Introduction (see Point 2). For example, you mention that you refrain from comparing results across datasets (line 362). That sounds a bit different in the Introduction and Results sections where you do compare results between datasets. Because the Introduction sets the stage for the Results, the major restrictions of the study, such as the small number of datasets and different "beauty" scores (see above), should be made clear already at the beginning of the paper.

Minor points

(8) Lines 223-227 and 251-253: The authors did not measure micro-experiences of fluency in their study. This is an interpretation that would be more appropriate for the Discussion section.

(9) Figures 1, 2, and S1: Some of the lettering is barely visible on a regular print-out.

(10) The present version of the code which the authors kindly provided at the Github site contains uncommented code from other experiments. A code restricted to the present experiments and annotated with more explanations would be helpful for users who want to understand the processing of the data in detail.

Reviewer #3: This is definitely an interesting article addressing the problem how to model beauty of different kinds, e.g. artworks, faces etc. The authors claim that neuronal sparsity correlates with beauty scores, explaining up to 28% of their variance and up to 47% when combined with a feature-based metric.

Their concept of sparsity is strongly referring to fluently processed visual stimuli. Later, they critically reflect on this model, which was put forward by the so-called hedonic fluency model, a very established model of aesthetic appreciation.

Importantly, they propose that it is not abvout the object as such, which is clever, but that beauty aspects arise by the interaction of the perceiver and the object; and fluency in this respect seems to be an ideal theoretical framework to model such interactive components. But here is also the fundamental problem of the highly appreciated approach provided by the authors: Fluency can, of course, be measured by the average speed of processed pictures, but much more interesting is the experimental treatment and manipulation of fluency and so how the neural network as a representative model would react on such increased familiarity and so fluency.

Besides this fundamental criticism, I really liked the general approach. I have found only the following aspects which should be addressed:

* Winkielman instead of Winkelman

* all statistical parameters should be italicized, e.g., p and R

* R2: 2 should be superscripted

* [42]: there exists an interesting addition: Carbon, C. C., & Albrecht, S. (2016). The Fluency Amplification Model supports the GANE principle of arousal enhancement. Behavioral and Brain Sciences, 39.

* beauty “as such”: must be made clear what kind of beauty aspects can be covered by the present approach, for instance, NOT extraordinary beauty

* other literature on different models should be included to assess the quality of predictive power of the current model against established ones

**Have the authors made all data and (if applicable) computational code underlying the findings in their manuscript fully available?**

Reviewer #1: **No: **The data used in this study are available from other articles (Refs. [24,25,26,27]). Code for replicating their findings is available in a git repository.

Reviewer #2: Yes

Reviewer #3: None

PLOS authors have the option to publish the peer review history of their article (what does this mean?). If published, this will include your full peer review and any attached files.

Reviewer #1: No

Reviewer #2: No

Reviewer #3: No
---

## [Decision Letter · Decision Letter 1]

20 Nov 2023

Dear Mr. Dibot,

We are pleased to inform you that your manuscript 'Sparsity in an artificial neural network predicts beauty: towards a model of processing-based aesthetics' has been provisionally accepted for publication in PLOS Computational Biology.

Best regards,

Roland W. Fleming, PhD

Academic Editor

PLOS Computational Biology

Natalia Komarova

Section Editor

PLOS Computational Biology

Reviewer's Responses to Questions

**Comments to the Authors: **

Reviewer #1: I have now studied the revised version of the manuscript "Sparsity in an Artificial Neural Network Predicts Beauty: Towards a Model of Processing-Based Aesthetics" by Dibot and co-authors, submitted for publication in PLOS Computational Biology.

I thank the authors for considering all my comments as well as for all the changes made in the manuscript. My comments were all properly addressed, and I am happy to reiterate my positive impression of this manuscript and to warmly recommend publication in the present form.

Reviewer #2: I thank the authors for addressing all my criticism and suggestions in great detail. I enjoyed the constructive exchange of scientific arguments during the reviewing procedure.

**Have the authors made all data and (if applicable) computational code underlying the findings in their manuscript fully available?**

Reviewer #1: Yes

Reviewer #2: Yes

PLOS authors have the option to publish the peer review history of their article (what does this mean?). If published, this will include your full peer review and any attached files.

Reviewer #1: No

Reviewer #2: No

---

## [Editor Report · Acceptance letter]

28 Nov 2023

PCOMPBIOL-D-23-00820R1 

Sparsity in an artificial neural network predicts beauty: towards a model of processing-based aesthetics

Dear Dr Dibot,

I am pleased to inform you that your manuscript has been formally accepted for publication in PLOS Computational Biology. Your manuscript is now with our production department and you will be notified of the publication date in due course.

With kind regards,

Dorothy Lannert
